A strength-oriented exercise session required more recovery time than a power-oriented exercise session with equal work

Helland Christian 1
Midttun Magnus 1
Saeland Fredrik 1
Haugvad Lars 1
Schäfer Olstad Daniela 2
Solberg Paul Andre 1
Paulsen Gøran goran.paulsen@nih.no 1 3
1 Norwegian Olympic and Paralympic Committee and Confederation of Sports , Oslo , Norway
2 Polar Electro Oy , Kempele , Finland
3 Department of Physical Performance, Norwegian School of Sport Sciences , Oslo , Norway
García-Ramos Amador
Electronic publication date: 2020 Sep 30
Publication date: 2020
Volume: 8
Electronic Location ID: e10044
Received 2020 Apr 16; Accepted 2020 Sep 4
Copyright: ©2020 Helland et al.
Copyright year: 2020
Copyright holder: Helland et al.
License: This is an open access article distributed under the terms of the Creative Commons Attribution License, which permits unrestricted use, distribution, reproduction and adaptation in any medium and for any purpose provided that it is properly attributed. For attribution, the original author(s), title, publication source (PeerJ) and either DOI or URL of the article must be cited.
License URL: https://creativecommons.org/licenses/by/4.0/

Keywords: Skeletal muscle, Athletes, Resistance exercise, Sprint running, Force-velocity, Countermovement jump, Squat jump, Rate of force development, Rate of perceived exertion, Perceived recovery status

Funding: The authors received no funding for this work.

==============================
The present randomized cross-over controlled study aimed to compare the rate of recovery from a strength-oriented exercise session vs. a power-oriented session with equal work. Sixteen strength-trained individuals conducted one strength-oriented session (five repetitions maximum (RM)) and one power-oriented session (50% of 5RM) in randomized order. Squat jump (SJ), countermovement jump (CMJ), 20-m sprint, and squat and bench press peak power and estimated 1RMs were combined with measures of rate of perceived exertion (RPE) and perceived recovery status (PRS), before, immediately after and 24 and 48 h after exercise. Both sessions induced trivial to moderate performance decrements in all variables. Small reductions in CMJ height were observed immediately after both the strength-oriented session (7 ± 6%) and power-oriented session (5 ± 5%). Between 24 and 48 h after both sessions CMJ and SJ heights and 20 m sprint were back to baseline. However, in contrast to the power-oriented session, recovery was not complete 48 h after the strength-oriented session, as indicated by greater impairments in CMJ eccentric and concentric peak forces, SJ rate of force development (RFD) and squat peak power. In agreement with the objective performance measurements, RPE and PRS ratings demonstrated that the strength-oriented session was experienced more strenuous than the power-oriented session. However, these subjective measurements agreed poorly with performance measurements at the individual level. In conclusion, we observed a larger degree of neuromuscular impairment and longer recovery times after a strength-oriented session than after a power-oriented session with equal total work, measured by both objective and subjective assessments. Nonetheless, most differences were small or trivial after either session. It appears necessary to combine several tests and within-test analyses (e.g., CMJ height, power and force) to reveal such differences. Objective and subjective assessments of fatigue and recovery cannot be used interchangeably; rather they should be combined to give a meaningful status for an individual in the days after a resistance exercise session.

Introduction

Athletes use different forms of resistance exercise (training) to improve muscle power output and sport performance, including heavy load strength-oriented exercises (e.g., ∼80% of 1 Repetition Maximum [1RM]) and low-to-moderate load power-oriented exercises (e.g., ∼40% of 1RM; Newton & Kraemer, 1994). In general, high-intensity resistance exercise challenges the ability to generate high forces, and with a conventional volume of exercise (∼5–15 sets per muscle group) neuromuscular fatigue develops during the sessions. This resistance exercise-induced fatigue typically requires 1–3 days of recovery (Vincent & Vincent, 1997; Raastad & Hallen, 2000; Ahtiainen et al., 2003; Ahtiainen et al., 2004).

The recovery process is obviously necessary for regaining full performance capacity, but it is also intertwined with adaptation processes, such as hypertrophy and increased efficacy of the metabolic pathways (Bishop, Jones & Woods, 2008; Paulsen et al., 2012; Cunanan et al., 2018). Recovery is therefore vital for all who perform resistance exercise, whether recreationally trained individuals or elite athletes. However, our knowledge of the recovery processes after resistance exercise is still inadequate, and we cannot accurately predict recovery times from a given exercise session (Bishop, Jones & Woods, 2008; Paulsen et al., 2012; Kellmann et al., 2018). The difficulty in predicting recovery rates lies in the range of factors at play, including—but not restricted to—type of muscle contractions, relative load (% of maximal strength) and volume of work performed (i.e., load times displacement times number of repetitions).

Contraction velocities and the transition from the eccentric to the concentric phase differentiate the diverse forms of resistance exercise. Classical power-oriented exercise means using low to moderate loads (e.g., 30–50% of 1RM) and the lifts are typically executed in a plyometric fashion; i.e., a fast transition from eccentric to concentric phase, and high (maximal effort) movement velocities (Newton & Kraemer, 1994; Suchomel, Comfort & Lake, 2017). This contrasts with strength-oriented exercise, in which the transition between the eccentric and concentric phase is typically controlled and slow (due to the heavy loads used).

Surprisingly, few studies have investigated the potential differences in recovery times between different forms of resistance exercise, such as strength-oriented exercise (>80% of 1RM) with slow velocities (mean velocity of the lifting bar <0.6 m/s) and power-oriented exercise with low/moderate loads (<50% of 1RM) lifted with high velocities (mean velocity of the lifting bar >1 m/s; Banyard et al., 2018; Garcia-Ramos et al., 2018b).

Linnamo, Hakkinen & Komi (1998) compared recovery rates after 40% of 10RM with 100% of 10RM (five sets and 2-minute rest periods) in the knee-extension exercise, applying a crossover design in non-resistance trained individuals. Using an isometric strength test, the authors demonstrated less acute fatigue and faster recovery from the power-oriented exercise compared to the heavy-load exercise over 48 h. Similarly, but studying elite track and field athletes, Howatson, Brandon & Hunter (2016) found a reduction in isometric strength 24 h after heavy-load strength-oriented session (4 × 5 repetitions; squat, split squat and push press), but not after power-oriented session (30% of the heavy load; 4 × 5 repetitions; speed squat, split squat jump and power press). However, with different exercise volumes (same total number of repetitions, but different loads), it is not possible to tease out the true impact of the load. Mccaulley et al. (2009) controlled for exercise volume and reported greater acute neuromuscular fatigue after heavy-load squats than after maximal power jump squats. However, there was no difference in the recovery between the strength-oriented session and power-oriented session after 24 and 48 h (Mccaulley et al., 2009). In a similar study, Hiscock et al. (2018) compared heavy loads (90% of 1RM; 3 × 3 reps) against “power loads” (45% of 1RM; 3 × 6 reps) in squat and deadlift with equal volume between exercise modes. No differences were found between experimental loads; however, recovery was seemingly complete within 12 h after the power-oriented session, while 24 h were required after the heavy-load session. In short, our knowledge of the impact of loads on recovery after different modes of resistance exercise is limited and requires further study.

Recovery can be defined as normalisation of neuromuscular function (Bishop, Jones & Woods, 2008). However, it is not always obvious which function(s) should be measured. In the Mccaulley et al. (2009) study, the participants conducted a dynamic squat exercise, but an isometric squat was used to assess neuromuscular function. Hence, it seems reasonable to question whether a dynamic test, such as squat jump (SJ) or countermovement jump (CMJ), would have displayed similar recovery rates. Indeed, when a range of recovery tests have been applied, such as CMJ, sprinting and single joint isokinetic torque, the tests do not demonstrate interchangeable recovery courses (Andersson et al., 2008; Chatzinikolaou et al., 2010). Apparently, different recovery rates may also be seen between variables extracted from the same test. For example, decrease in mean power has been shown to recover faster than the increased duration of the concentric phase of the CMJ (Gathercole et al., 2015a).

To confidently track the time course of recovery, dealing with the error of measurements is a challenge. Impairments of muscle strength and power are typically observed in the range of ∼5–20% immediately after resistance exercise in trained individuals, but may be less than 5% below baseline after 24 h (Raastad & Hallen, 2000; Howatson, Brandon & Hunter, 2016; Hiscock et al., 2018). Knowing that the typical error (coefficient of variation; CV) in day-to-day measurements of CMJ and SJ height and power is at best ∼3–5% (Raastad & Hallen, 2000; Hopkins, Schabort & Hawley, 2001; Gathercole et al., 2015a), the sensitivity of vertical jump tests may be limited for monitoring the final part of the recovery process. In the present study we address the typical error of all tests applied and explore the sensitivity of different variables extracted from SJ and CMJ (including jump height, peak power and peak force).

Exercise load and work, neuromuscular fatigue and recovery can be assessed not only with objective performance measures (strength and power tests; as discussed above), but also subjectively, using rate of perceived exertion (RPE) and perceived recovery status (PRS). Perceived exertion has been used for years with endurance exercise (Borg, 1970), but also for resistance exercise (Foster, Rodriguez-Marroyo & De Koning, 2017), while the PRS scale has a shorter history (Laurent et al., 2011). Subjective measures are simple and convenient to use, but what do they tell us? The use of and correlation between objective and subjective measures of recovery and performance monitoring have been debated for years (Scott et al., 2016; Foster, Rodriguez-Marroyo & De Koning, 2017).

Interestingly, few investigations have compared subjective and objective recovery assessments after different forms of resistance exercise. Sikorski et al. (2013) observed a relationship between PRS and the muscle damage marker creatine kinase 48 h after a conventional, high-volume resistance exercise session; however, no measure of muscle function was included. Korak, Green & O’neal (2015) observed that recreationally strength-trained males experienced faster recovery, measured with PRS, from single-joint compared to multi-joint exercises, which appeared to correspond to changes in 10RM-tests (objective tests). Unfortunately, the study lacked appropriate objective measures of recovery; i.e., tests of maximal force and power. In a case study of three weightlifters/powerlifters, Zourdos et al. (2016) found that daily 1RM lifts consistently improved performance over 37 days, but the improvement seemed inadequately reflected in RPE and PRS scores. Although case studies of high-level athletes are interesting, such studies provide only weak evidence. This leaves us with the conclusion that more research is needed to elucidate the relationship between objective and subjective measures of recovery after different forms of resistance exercise.

The aim of the present study was to compare the recovery rates from a power-oriented session with a heavy-load strength-oriented session of similar work. A range of objective performance tests of strength and power were combined with subjective tests (RPE and PRS) to acquire a broad picture of the recovery processes in both upper and lower body muscles. We hypothesized that the power-oriented session would induce less performance decrements than the strength-oriented session at all time points; and, consequently, that complete recovery would occur within 48 h for the power-oriented session but not the strength-oriented session. Secondly, compared to the strength-oriented session, we hypothesized that the participants would perceive the power-oriented session as less strenuous and to experience a better recovery status after 24 and 48 h. Finally, we hypothesized that changes in RPE and PRS would be numerically related to changes in objective measures, such as the SJ and CMJ.

Materials & Methods

Study design

The present study was a randomized cross-over study where each participant completed a heavy-load, strength-oriented session and a moderate-load, power-oriented session, in randomized order (applying the Research Randomizer; Urbaniak & Plous, 2013). To achieve a counterbalanced order, the participants were paired, so that one started with the strength-oriented session and one with the power-oriented session. One to four weeks of rest were allowed between sessions (16 ± 10 days (mean ± standard deviation)).

A test battery of physical performance tests and evaluation of perceived effort and recovery status was applied before, immediately after, and 24 and 48 h after the exercise sessions (Fig. 1). The concentric work (J) performed in the first session was recorded and replicated in the second session, ensuring equal work in both sessions. The exercises were the same for both sessions, but somewhat adapted to serve the purpose of the sessions (Table 1). The primary aim of the study was to compare the recovery rates between sessions when all factors were equal except the external load (50% lower in the power-oriented session than the strength-oriented session).

Figure 1 Overview of the study design.

The session that was performed first, either the strength-oriented session or the power-oriented session, was randomized.

Table 1 The exercises applied.

Exercises for each of the two sessions.

Power session	Heavy strength session	Comment	
Loaded CMJ	Squat	Same depth in the eccentric phase	
Front squat with overhead push	Front squat	Same depth in the eccentric phase	
Trap bar CMJ	Trap bar squat	Same depth in the eccentric phase	
Bench press throw	Bench press	Conducted in a Smith rack	
Narrow bench press throw	Narrow bench press	Conducted in a Smith rack	
Explosive push-ups	Weighted push-ups	Load by weight-vest (1–9 kg) and discs (5–20 kg). Boxes (25 cm) were placed under feet and hands.	

Three to seven days before the first exercise session, a familiarization session was conducted. The participants were familiarized with all tests and exercises (see details below) and instructed not to conduct any strenuous exercise 48 h prior to the test days. The participants were also instructed to standardize their breakfast before and their meals after the exercise sessions (for 48 h). All supplements and medications were prohibited during the study period.

During the exercise sessions the participants were given a protein bar and a protein drink (both supplements containing approximately 20 g protein, 30 g carbohydrates, and a total of ∼1,000 kJ (Yt, Tine, Oslo, Norway), and an energy drink (30 g carbohydrates; 510 kJ; Yt, Tine, Oslo, Norway) to ensure sufficient protein and energy intake (in total: 40 g protein and 90 g carbohydrates; ∼1,500 kJ). Water was allowed ad libitum.

Participants

Nineteen young, resistance-trained individuals were recruited to this study. Sixteen participants, eight males and eight females, completed all tests and both exercise sessions (24 ± 3 years, 74 ± 12 kg, 1.75 ± 0.11 m; Table 2). Two participants dropped out due to muscle pains (hamstrings and groin) during testing or the exercise sessions; and one was excluded due to technical problems with the test equipment.

Table 2 Baseline values for the strength-oriented session and the power-oriented session.

Variable	PowerMean ± SD	StrengthMean ± SD	SD used for standardizing(adjusted)	Smallest worthwhile change % (0.2 SD)	Coefficient of Variation % (CV)	
CMJ (cm)	34.8 ± 8.7	34.7 ± 9.0	8.9	5.0	5.1	
CMJ peak power (W)	1905 ± 670	1869 ± 722	703	7.5	6.5	
CMJ mean power (W)	316 ± 116	317 ± 128	123	8.0	8.7	
CMJ concentric peak force (N)	1788 ± 406	1774 ± 348	381	4.3	4.0	
CMJ RFDmax (N/s)	13169 ± 5317	12843 ± 5895	5663	8.8	21.2	
CMJ duration (s)	0.84 ± 0.08	0.84 ± 0.09	0.09	2.1	7.4	
CMJ eccentric peak force (N)	1793 ± 410	1787 ± 357	378	4.4	4.2	
CMJ eccentric time (s)	0.18 ± 0.04	0.19 ± 0.03	0.04	3.8	9.9	
CMJ depth (cm)	39.2 ± 6.0	40.1 ± 6.4	6.3	3.2	8.3	
SJ (cm)	32.0 ± 8.0	32.3 ± 8.2	8.2	5.0	5.7	
SJ peak power (W)	1980 ± 672	2003 ± 748	717	7.3	6.3	
SJ mean power (W)	586 ± 220	606 ± 254	240	8.2	9.8	
SJ peak force (N)	1630 ± 326	1637 ± 361	347	4.3	4.3	
SJ RFDmax (N/s)	7155 ± 2090	7675 ± 3210	2744	6.9	21.0	
SJ duration (s)	0.40 ± 0.03	0.40 ± 0.05	0.04	2.1	8.6	
MJ (cm)	27.6 ± 6.8	29.6 ± 8.2	7.7	5.2	9.1	
MJ RSI	45.2 ± 12.0	47.4 ± 15.6	14.1	6.0	14.9	
MJ vertical stiffness (N/m)	6.0 ± 1.9	5.9 ± 1.7	1.8	6.0	19.9	
20 m (s)	3.08 ± 0.22	3.08 ± 0.23	0.23	1.5	1.3	
Push-up peak force (N)	1071 ± 421	1105 ± 422	425	7.1	11.2	
Squat peak power (W)	1380 ± 332	1438 ± 314	327	4.8	7.2	
Bench press peak power (W)	433 ± 180	450 ± 184	184	9.4	9.3	
Squat estimated 1RM (kg)	121 ± 39	120 ± 41	39.9	6.6	4.6	
Bench press estimated 1RM (kg)	80 ± 29	81 ± 30	30.0	7.5	3.3	
PRS whole body (0–100)	83.1 ± 9.5	76.9 ± 10.1	10.4	10	14.5	
Total work upper body (kJ)	12 ± 7	11 ± 7	–	–	–	
Total work lower body (kJ)	57 ± 14	57 ± 14	–	–	–	
Notes.

1RM 1 Repetition Maximum

CMJ Countermovement Jump

MJ Multi Jump

PRS Perceived Recovery Status

RSI Reactive Strength Index

RFDmax Maximal Rate of Force Development

SJ Squat Jump

The participants were familiar with heavy-load strength training and had been training upper and lower body strength exercises on a weekly basis during the last year (≥2 sessions/week). Of the 16 participants, three were competing at a national elite level (two volleyball and one beach volleyball player), one was an international-level bike trial athlete, and the remaining 12 participants were students at the Norwegian School of Sport Sciences (Oslo, Norway) and engaged in strength training at a recreational level.

The study was reviewed by the Norwegian Regional Ethical Committee of Medical and Health Research (2016/1120). The participants gave written informed consent to take part in the study, following the Declaration of Helsinki (World Medical Association).

Testing and exercises

The familiarization session consisted of all the tests (see below) and 1–3 sets of five repetitions of all the exercises (for both sessions): squat, front squat, trap bar squat, bench press, narrow bench press and push-ups (Fig. 1). The loads were adjusted to get close to a 5-repetition maximum (RM) during the last set. For the power exercises the loads were 50% of the estimated 5RM loads. In both sessions, the exercises were executed with maximal effort in the concentric phase in all repetitions. In the strength-oriented session, the eccentric phase was conducted with a controlled, slow movement (>1 s). In contrast, in the power-oriented session the eccentric phase was faster (<1 s) in order to maximize the power output in the concentric phase. The movement velocity was measured using a linear encoder (see description below).

On the days of the exercise sessions, the participants rated their perceived recovery status (PRS scale; 0–10; Laurent et al., 2011) prior to a warm-up. The warm-up consisted of a 10-minute easy run with increasing velocity (not more than moderate effort) preceding 2 min of individually selected dynamic stretching of both upper and lower body muscles. Thus, each participant followed a warm-up procedure that they were accustomed to, and the procedure was similar before each session (including the familiarization session).

The tests were then conducted in the following order: CMJ, SJ, 10 consecutive multiple jumps (MJ), 20-meter sprint running, maximal push-up force, and power profiles and estimated 1RMs in bench press and squat. Tests were performed before and immediately after the sessions, and again after 24 and 48 h. The power profile tests and 1RM estimation in the bench press and squat were, however, not conducted immediately after the sessions in order to prevent additional fatigue. Finally, 30 min after the sessions the participants rated the perceived exertion (session RPE; 0–10; Foster et al., 2001). The participants were introduced to the ratings and descriptors of both the RPE and the PRS scales at the familiarization session.

Tests

The countermovement jump (CMJ), squat jump (SJ), and multi-jump (MJ) were conducted on an AMTI force platform (sampling rate: 2000 Hz; OR6-5-1; AMTI, Watertown, MA, USA). Before every test session (for each participant), offset values were acquired and the body weight of the participant was measured and averaged over a 2.3 s period (as recommended by Street et al., 2001). The body weight measured before each jump was confirmed against the initial weigh-in value (less than 5% discrepancy was considered valid). All force data were filtered with a low pass filter (second order Butterworth bi-directional low pass filter; cut-off frequency of 120 Hz).

All jump-tests were performed with the hands fixed on the hips (akimbo). Based on jump height, the average of each individual’s two best attempts of 3–6 jumps was used for subsequent statistical analyses—except for MJ, where only one attempt was made (due to the development of fatigue). The inter-test coefficients of variation (CVs) for the jump tests are listed in Table 2.

All data collected from the AMTI force plate were analysed using a custom-made software (Matlab, The Mathworks, Inc., MA, USA; Biomekanikk AS, Oslo, Norway). From the SJ, duration, concentric peak force, peak and mean power, RFDmax, and jump height were calculated. From the CMJ, duration, eccentric time, peak and mean eccentric and concentric force, RFDmax, and jump depth (lowering of the center of mass (COM)) and jump height were calculated. From the MJ, jump height, vertical stiffness, and reactive strength index (RSI) were calculated.

Jump height was calculated as the squared take-off velocity divided by 2 g for all jumps (SJ, CMJ, and MJ). Take-off velocity was calculated by the impulse–momentum method described by Linthorne (2001) and Street et al. (2001), with the impulse-integral starting from the time point when vertical force exceeded (or fell below for CMJ) 100% of body weight and ending when force fell below 2 N (take-off).

The jump’s phases were calculated as follows: Duration (s) of the SJ was found by backtracking force data from take-off (force <2 N) to the point where the force was 101.5% of body weight. CMJ was divided into an eccentric phase and a concentric phase (Fig. 2), defined by the phase where the COM was descending and ascending, respectively. The initiation of the CMJ (eccentric phase) was found by backtracking the force data from the point of zero velocity, i.e., the deepest position of the COM, to the point where the force was 98.5% of body weight. Eccentric peak force was the highest force measured within the eccentric phase, and eccentric time was defined as the duration of the eccentric phase where the force was greater than that of the body weight (Fig. 2). Peak concentric force was the highest force measured from the point of zero velocity of the COM to the point of take-off.

Figure 2 Force-time curve of a countermovement jump.

An example of a force-time curve of a countermovement jump (CMJ). The eccentric and concentric phase are displayed. RFD, Rate of Force Development.

The maximal rate of force development (RFD) was defined as the largest increase in force over a 5 ms time window during the jump (both for SJ and CMJ; Fig. 2). Specifically, the RFD values (N/s) were calculated from numerical differentiation of the low-pass filtered force measurements using a 4-point method, and the derivative was averaged over 5 ms (10 samples). In the MJ test, the participants were instructed to jump ten consecutive CMJs as high as possible. The vertical stiffness was calculated as the maximal force divided by the downward displacement of the COM, while the reactive strength index (RSI) was calculated as jump-height divided by the ground contact time. All variables are presented as the average of the ten jumps.

Two to three maximal 20-meter sprint runs were performed on a rubberized indoor track (Mondo, Conshohocken, PA, USA) with 3–4 minutes’ rest between trials. The sprints were measured with an electric timing system (Biomekanikk AS, Oslo, Norway) with a timing trigger (single-beamed timing gate 0.6 m after the start line and 0.4 m above ground level) and dual-beamed timing gates placed every 5 m along the sprint track. Participants were instructed to accelerate as fast as possible from a standing start with one foot in front of the other. The inter-test CV for the sprint test is given in Table 2.

After a specific warm-up consisting of ten push-ups with gradually increasing effort and three maximal singles, three single maximal push-ups were assessed on a force platform (sampling rate: 2,000 Hz; OR6-5-1; AMTI, Watertown, MA). One minute of rest was given between the single push-up efforts. The participants were instructed to keep their body “straight” (minimize any movement of the spine and pelvis) and to do a controlled slow eccentric phase to a position where the chest was 2–3 cm above the floor, and then do a push as fast as possible. The hands were allowed to leave the platform if the push-force was large enough to lift the upper body off the ground (the feet were always in contact with the ground). Hand and foot placements were standardized for each participant. If the participants failed to conduct the push as described, the attempt was discarded and repeated (this was, however, a subjective decision by the test leader). The inter-test CV for the push-up test is given in Table 2.

Bench press and squat performance were assessed using a linear encoder (Musclelab Linear Encoder; Ergotest Innovation, Langesund, Norway). The string of the encoder was attached to the bar, with the device measuring displacement (d) and time of the concentric phase (200 Hz sampling rate; 0.019 mm resolution). The start and end of the concentric phase were detected as a 5 ms period of no movement (<0.004 m/s) or immediately by a change in direction (within 5 ms). The calculations of velocity and force from each load were based on the entire concentric phase (i.e., average/mean velocity and force; v = d/t; acceleration [a] = v/t, force [F] = mg + ma).

In both the bench press and squat the participants completed sets of three maximal repetitions at four different loads, with ∼5 s between each lift and 2–4 min between sets. All repetitions were conducted with maximal effort in the concentric phase. The external loads were 25, 50, 75 and 90% of estimated 1RM (estimated during the familiarization session). The attempts with the highest velocity from each load were selected for further analysis. A concentric force-velocity relationship (linear regression) and a power-velocity relationship (parabolic curve) were established and peak power and 1RM were estimated (software from Ergotest Innovation, Langesund, Norway). Peak power was calculated as the apex of the parabolic power-velocity relationship. The 1RM was calculated from the intercept of the load (mg) − velocity relationship and the force (mg + ma) − velocity relationship.

For the squat, the participants were instructed to squat down to a position where the femur was parallel with the floor, in a slow, controlled manner, and then extend as rapidly and powerfully as possible. For the squat we estimated force from the system mass (90% of body mass and the external mass), while for the bench press, only the external mass was used. We used 90% of the body weight for the squat calculations, as suggested by the manufacturer (Ergotest Innovation, Langesund, Norway). This is very close to the 88% of body weight suggested by others using a similar linear encoder device (Cormie, Mcbride & Mccaulley, 2007). The inter-test CVs for the bench press and squat tests are given in Table 2.

Exercise sessions

The strength-oriented session consisted of three exercises for the lower body, in the following order: squat, front squat, trap bar squat; and three exercises for the upper body, performed in the following order: normal bench press, narrow bench press and weighted push-ups (Table 1). A warm-up set of 8 repetitions at 60–80% of 5RM before each exercise preceded 5 sets of 5RM. The 5RM loads were estimated from the familiarization session for each exercise. The inter-set rest period was 3–4 min. The loads were adjusted between sets, if necessary. All exercises were conducted at the same tempo with a controlled slow eccentric phase and a fast as possible concentric phase. The leg exercises were performed with free weights (Eleiko, Halmstad, Sweden), while both normal and narrow bench press exercises were performed in a Smith rack (Multipower, Technogym, Cesena FC, Italy). Weighted push-ups were performed on three 30 cm custom-made boxes, and loads were applied by a weight-vest (1–9 kg; Reebok, Boston, Ma, US) and (if needed) weight discs (5–20 kg) placed on the participant’s back, positioned over the scapulae.

The power-oriented session was conducted with loads corresponding to 50% of the external load used in the strength-oriented session. Loaded CMJ, front squat with overhead push, trap bar CMJ, normal bench press throw (Smith rack), narrow bench press throw (Smith rack), and explosive push-ups were performed with a continuous high velocity tempo in the concentric phase (Table 1).

We measured the concentric displacement for all the exercises in both sessions with a linear encoder (see above). The encoder’s string was attached to the bar in all cases except for both push-up variations, where the string was attached to a light chest belt at the distal part of the sternum.

The total work was calculated by summarizing the products of repetitions, load and displacement for each set of each exercise (Table 2). Only the displacement of the concentric phase was used; i.e., the distance from the vertically lowest to the vertically highest position of the bar in the squat exercise. For the lower body exercises we assumed the load to be the sum of 90% of the body weight and the external load (see above). For the front squat push, the squat part was calculated as a regular squat, but for the final overhead push only the external load was used; thus, the squat work and push work were calculated separately and then added together. For the bench press exercises, only the external load was used, while for the push-ups the weight of the upper body (measured with the force plate during testing) was added to the external load.

The first session (randomly strength or power) was used as a template for the second session for each participant. Hence, we adjusted the number of sets per exercise so that the concentric work performed in each exercise was similar between sessions. The amount of work per exercise was fine-tuned by adjusting the number of repetitions in the final set (e.g., performing only two repetitions in order to reach the required amount of work).

Statistics

The data were analysed in spreadsheets that enabled adjustment of one or two predictor variables in the changes within or difference between sessions (Hopkins, 2007). The spreadsheet is fundamentally based on the T-test but gives the opportunity to adjust for baseline to control for the regression to the mean effect. All data were log-transformed, and changes are reported as percentages with their associated 95% confidence interval (CI).

The reliability of the tests was based on the familiarization session and the two pre-tests (before each session). The coefficient of variation (CV) was calculated as described by Hopkins (2000). The smallest worthwhile change was calculated as the baseline between subjects’ standard deviation (SD) multiplied by 0.2 (Hopkins, 2004).

Effects were evaluated using clinical magnitude-based inferences (MBD) (Hopkins et al., 2009; Hopkins, 2019), a method appropriate for small samples. The magnitude of changes within and difference in mean between sessions were assessed by standardization (mean change/difference divided by baseline SD of all subjects), and the resulting standardized effect evaluated with a modification of Cohen’s (1992) scale: <0.2, trivial; 0.2–0.6, small; 0.6–1.2, moderate; >1.2, large (Hopkins et al., 2009). The subjective variables (RPE and PRS) were evaluated with the following scale: <10% trivial, 10–30% small, 30–50% moderate, 50–70% large, 70–90% very large, and 90–100% extremely large (Hopkins, 2010).The initial RPE and PRS values were therefore factored by 10 (0–100).

To make clinical inferences about true values of effects in the population studied, the effects were expressed as probabilities of harm or benefit in relation to the smallest worthwhile change (0.2 of SD; Hopkins et al., 2009). A clear change within or difference between the two exercise modalities corresponds to the case of an effect that is almost certainly not harmful (<0.5% risk of harm) and possibly beneficial (>25% chance of benefit). The effect is shown as the difference or change with the greatest probability, and it is shown qualitatively using the following scale: 25–75%, possibly; 75–95%, likely; 95–99.5%, very likely; >99.5%, most likely (Hopkins et al., 2009). In addition, p-values were included, and effects were considered significant at p < 0.05 if the 95% CI did not overlap zero (p < 0.01 with 99% CI).

Correlations between variables were obtained using Pearson’s r (and 95% CI). We restricted the correlation analyses to testing between objective and subjective variables that showed a difference between sessions, in order to minimize the risk of observing random correlations.

An order-effect is a potential risk with a crossover design (Woods, Williams & Tavel, 1989). Hence, we tested the session-order effect by including session order as a covariate. The effects were trivial (0.0–0.5%), so we did not further include the order effect to avoid too many covariates with the relatively low sample size (baseline value was already included).

Results

Baseline values for the 16 participants are presented in Table 2. The differences between the two modalities at baseline were all trivial; nevertheless, baseline values were included as a covariate in all analyses of within-session changes and between session differences, and thereby controlled for.

The smallest worthwhile change (SWC) and the CV for each variable are presented as relative values (Table 2). Note that the CV was larger than the SWC for most variables (e.g., CMJ and SJ RFDmax), but equal or lower for some variables (e.g., eccentric peak force).

Figure 3 Variables derived from the countermovement jump (CMJ) test obtained before, immediately after (0 hours) and 24 and 48 hours after the strength-oriented session and the power-oriented session.

Values are means and 95% CIs of percentage changes from pre-values. Changes within sessions and differences (Diff) between sessions are marked with effect sizes and p-values. Grey areas represent the smallest worthwhile change. (A) Jump height, (B) Peak power, (C) Mean power, (D) Peak concentric force, (E) RFDmax, (F) Eccentric time, (G) Eccentric peak force, (H) Depth (lowering of center of mass). RFDmax, Maximal Rate of Force Development. Trivial (Triv): <0.2, Small: 0.2–0.6; Moderate (Mod): 0.6–1.2; Large: 1.2–2.0; Very large: 2.0–4.0; Extremely large: <4.0 *: Possibly beneficial, **: Likely beneficial, ***: Very likely beneficial +: Possibly harmful, ++: Likely harmful, +++: Very likely harmful, ++++: Most likely harmful 0: Possibly trivial, 00: Likely trivial, 000: Very likely trivial, 0000: Most likely trivial Uncl: Unclear a: p < 0.05 b: p < 0.01.

Within-session changes immediately after (0 h), and 24 and 48 h after the sessions are shown in Fig. 3 and Table 3. Immediately after the sessions the changes were generally negative: both sessions showed small clear negative changes for most CMJ variables (height, mean power, concentric peak force, eccentric peak force; Fig. 3) and SJ mean power (Table 3). The CMJ RFDmax and the subjective PRS variable had a clear moderate negative change after both sessions. In addition, the strength-oriented session gave clear small negative changes in CMJ depth, SJ height, SJ RFDmax, SJ duration and MJ RSI, while these were trivial after the power-oriented session (Table 3 and Fig. 3). The sRPE values (0–100) for the power-oriented session were 50 ± 13, 51 ± 15 and 53 ± 11 for whole body, upper body and lower body, respectively; and correspondingly, 71 ± 15 (whole body), 68 ± 15 (upper body) and 74 ± 13 (lower body) for the strength-oriented session.

At 24 h similar trends emerged, with the strength-oriented session showing clear small negative effects on CMJ peak concentric force (Fig. 3), SJ RFDmax, SJ duration and squat peak power; while these changes were trivial after the power-oriented session (Table 3). In addition, the strength-oriented session showed a clear moderate negative effect on CMJ eccentric time (Fig. 3) and total and lower body PRS, compared to a small negative effect after the power-oriented session. In contrast, the power-oriented session gave a small possibly beneficial effect on MJ height. MJ vertical stiffness had a small increase after the strength-oriented session, while it had a clear decrease after the power-oriented session.

At 48 h, most clear negative changes were small and only evident after the strength-oriented session (Table 3 and Fig. 3). Further, CMJ RFDmax and CMJ eccentric time displayed clear moderate negative changes after the strength-oriented session (Fig. 3); this was also reflected in a small increase in total duration of the CMJ (5.3 ± 3.5%) 48 h after the strength-oriented session. In contrast to the strength-oriented session, the power-oriented session resulted in a small possibly beneficial change in squat peak power at 48 h (Table 3).

A few clear differences were observed between sessions (Fig. 3 and Table 4). Compared to the power-oriented session, the strength-oriented session showed small negative effects on CMJ depth, SJ duration and MJ height immediately after the session. Moreover, the strength-oriented session was rated higher on the sRPE scale than the power-oriented session (small effect). At 24 h, the strength-oriented session showed small clear negative effects on CMJ depth and eccentric peak force, SJ RFDmax, MJ height, squat peak power, and total, upper and lower body PRS compared to the power-oriented session. On the other hand, the strength-oriented session had a small and likely beneficial effect on MJ vertical stiffness compared to the power-oriented session.

At 48 h, the strength-oriented session still demonstrated small and possibly to likely negative effects compared to the power-oriented session for CMJ concentric and eccentric peak forces, SJ RFDmax, push-up peak force and upper body PRS. The differences between sessions in CMJ depth and squat peak power were partly due to improvements over baseline after the power-oriented session.

To investigate the relationship between subjective and objective tests, we selected the objective tests that demonstrated the greatest difference between the sessions. Hence, we correlated the CMJ eccentric peak force against PRS at 24 and 48 h after exercise; and, for the upper body, push-up peak force against PRS at 24 and 48 h after exercise (Fig. 4). There were no clear positive or systematic correlations between these variables. There was a clear negative correlation between push-up peak force and PRS at 24 h after the power-oriented session (but not after 48 h), indicating a counterintuitive relationship between high force (i.e., indicating a high degree of recovery) and a low degree of perceived recovery.

Table 3 Changes and recovery over time.

Percent changes from baseline within the strength-oriented session and the power-oriented session, with 95% CIs and associated effect sizes and inferences (adjusted for baseline values).

Variable		Post 00 hoursMean ± SD;±95%CI	Inference	Post 124 hoursMean ± SD; ±95%CI	Inference	Post 248 hoursMean ± SD; ±95%CI	Inference	
SJ height	Power	−4.2 ± 3.8; ±1.9	Triv00 (p < .05)	−1.2 ± 3.9; ±2.1	Triv0000	0.7 ± 4.4; ±2.4	Triv0000	
	Strength	−8.2 ± 5.8; ±2.8	Small+++ (p < .05∕01)	−3.7 ± 6.5; ±3.3	Triv00 (p < .05)	−2.1 ± 6.7; ±3.4	Triv000	
SJ peak power	Power	−2.9 ± 4.3; ±2.2	Triv000 (p < .05)	−1.5 ± 5.6; ±2.9	Triv000	−1.1 ± 7.0; ±3.7	Triv000	
	Strength	−4.3 ± 4.8; ±2.4	Triv+ (p < .05∕01)	−2.6 ± 9.4; ±4.7	Triv00	−3.8 ± 8.6; ±4.3	Triv+	
SJ mean power	Power	−5.9 ± 7.9; ±3.8	Small+ (p < .05∕01)	−5.4 ± 11.8; ±5.7	Small+	−1.5 ± 12.9; ±6.7	Trivuncl	
	Strength	−11.5 ± 12.9; ±5.8	Small+++ (p < .05∕01)	−7.8 ± 14.1; ±6.5	Small++ (p < .05)	−6.3 ± 15.1; ±7.1	Small+	
SJ peak force	Power	−0.7 ± 3.2; ±1.7	Triv0000	−0.9 ± 3.7; ±1.9	Triv0000	−1.4 ± 4.3; ±2.3	Triv0000	
	Strength	−0.6 ± 3.0; ±1.6	Triv0000	−1.2 ± 5.8; ±3.0	Triv000	−2.7 ± 4.6; ±2.4	Triv000 (p < .05)	
SJ RFDmaks	Power	−4.4 ± 15.3; ±7.3	Triv+	0.0 ± 16.0; ±7.9	Triv00	4.2 ± 17.5; ±9.4	Trivuncl	
	Strength	−7.0 ± 17.3; ±8.0	Small+	−11.5 ± 28.1; ±11.8	Small++	−7.3 ± 32.9; ±14.2	Small+	
SJ duration	Power	1.9 ± 6.8; ±3.6	Triv000	4.1 ± 9.8; ±5.2	Triv+	1.7 ± 9.5; ±5.1	Triv00	
	Strength	5.3 ± 8.8; ±4.8	Small+ (p < .05)	5.5 ± 9.8; ±5.3	Small+ (p < .05)	4.4 ± 9.8; ±5.2	Triv+	
MJ heigth	Power	2.4 ± 12.4; ±6.4	Trivuncl	6.4 ± 9.3; ±5.3	Small* (p < .05)	4.4 ± 10.7; ±5.7	Triv*	
	Strength	−3.2 ± 10.0; ±4.1	Triv00	−0.5 ± 8.4; ±3.5	Triv000	1.4 ± 4.8; ±2.1	Triv0000	
MJ RSI	Power	−2.5 ± 14.7; ±7.2	Triv00	4.1 ± 11.3; ±6.2	Smalluncl	1.1 ± 11.3; ±5.8	Triv00	
	Strength	−6.3 ± 11.4; ±5.4	Small+ (p < .05)	−0.7 ± 7.3; ±3.8	Triv000	1.6 ± 8.3; ±4.4	Triv000	
MJ vertical stiffness	Power	2.8 ± 19.9; ±10.4	Trivuncl	−5.3 ± 18.0; ±9.1	Small+	−5.2 ± 24.4; ±11.6	Small+	
	Strength	0.4 ± 11.6; ±6.2	Triv00	5.3 ± 10.5; ±5.8	Small*	−1.1 ± 15.2; ±7.8	Triv00	
20 m	Power	0.0 ± 1.8; ±1.0	Triv0000	0.7 ± 1.6; ±0.8	Triv000	0.6 ± 1.7; ±0.9	Triv000	
	Strength	1.5 ± 1.8; ±1.0	Triv+ (p < .05)	1.2 ± 1.9; ±1.1	Triv+ (p < .05)	0.5 ± 1.7; ±1.0	Triv000	
Push-up peak force	Power	−0.6 ± 12.0; ±6.1	Triv000	−4.8 ± 14.4; ±7.5	Triv+	5.2 ± 14.4; ±8.6	Trivuncl	
	Strength	−1.0 ± 7.3; ±3.1	Triv0000	−4.7 ± 9.6; ±3.9	Triv00	−4.3 ± 7.1; ±2.9	Triv00	
Squat peak power	Power	–	–	2.9 ± 9.0; ±4.7	Triv00	5.8 ± 6.7; ±3.8	Small* (p < .05)	
	Strength	–	–	−6.4 ± 7.6; ±3.7	Small++ (p < .05∕01)	−3.7 ± 8.3; ±4.7	Triv+	
Bench press peak power	Power	–	–	−0.1 ± 5.3; ±3.0	Triv0000	3.5 ± 8.9; ±5.2	Triv000	
	Strength	–	–	−5.6 ± 6.3; ±3.4	Triv000(p < .05∕01)	−2.6 ± 10.7; ±6.4	Triv000	
Squat estimated 1RM	Power	–	–	−1.6 ± 6.2; ±3.2	Triv0000	−1.3 ± 5.5; ±2.9	Triv0000	
	Strength	–	–	−0.5 ± 5.7; ±3.0	Triv0000	−2.5 ± 4.8; ±2.8	Triv0000	
Bench press estimated 1RM	Power	–	–	−1.6 ± 4.8; ±2.7	Triv0000	−1.1 ± 3.9; ±2.2	Triv0000	
	Strength	–	–	−3.2 ± 5.4; ±3.0	Triv000(p < .05)	−2.8 ± 5.0; ±3.1	Triv0000	
PRS whole body	Power	−36.9 ± 16.5; ±8.9	Mod++++ (p < .05∕01)	−23.8 ± 8.7; ±4.7 (p < .05∕01)	Small++++	−13.6 ± 9.1; ±4.7 (p < .05∕01)	Small++	
	Strength	−44.1 ± 15.6; ±8.4	Mod++++ (p < .05∕01)	−30.0 ± 11.9; ±6.4 (p < .05∕01)	Mod++++	−16.3 ± 14.7; ±7.9 (p < .05∕01)	Small+++	
PRS upper body	Power	−38.1 ± 17.5; ±9.4	Mod++++ (p < .05∕01)	−21.3 ± 10.4; ±5.6 (p < .05∕01)	Small++++	−11.3 ± 8.8; ±4.7 (p < .05∕01)	Small+	
	Strength	−43.8 ± 11.7; ±6.2	Mod++++ (p < .05∕01)	−29.4 ± 10.3; ±5.5 (p < .05∕01)	Small++++	−17.5 ± 18.1; ±9.7 (p < .05∕01)	Small++	
PRS lower body	Power	−40.6 ± 14.5; ±7.8	Mod++++ (p < .05∕01)	−24.4 ± 11.3; ±6.0 (p < .05∕01)	Small++++	−15.6 ± 13.0; ±7.0 (p < .05∕01)	Small++	
	Strength	−45.0 ± 13.6; ±7.3	Mod++++ (p < .05∕01)	−32.5 ± 13.3; ±7.1 (p < .05∕01)	Mod++++	−16.9 ± 12.5; ±6.7 (p < .05∕01)	Small+++	
Notes.

1RM 1 Repetition Maximum

CI Confidence Interval

MJ Multi Jump

PRS Perceived Recovery Status

RSI Reactive Strength Index

RFDmax Maximal Rate of Force Development

SD Standard Deviation

SJ Squat Jump

Trivial (Triv): <0.2, Small: 0.2-0.6; Moderate (Mod): 0.6-1.2; Large: 1.2-2.0; Very large: 2.0-4.0; Extremely large: <4.0 *: Possibly beneficial, **: Likely beneficial, ***: Very likely beneficial +: Possibly harmful, ++: Likely harmful, +++: Very likely harmful, ++++: most likely harmful 0: Possibly trivial, 00: Likely trivial, 000: Very likely trivial, 0000: Most likely trivial uncl: Unclear (need more data) p < .05: The 95% CI do not overlap with zero p < .01: The 99% CI do not overlap with zero.

Table 4 Differences between sessions.

Percent differences between the strength-oriented session and the power-oriented session at 0, 24 and 48 hours, with 95% CIs and associated effect sizes and inferences (strength minus power; adjusted for baseline values).

Variable	Post 00 hMean; ±95% CI	Inference	Post 124 hMean; ±95% CI	Inference	Post 248 hMean; ±95% CI	Inference	
SJ height	−4.1; ±3.4	Triv+ (p < .05)	−2.5; ±3.8	Triv00	−2.8; ±3.6	Triv00	
SJ peak power	−1.4; ±3.5	Triv0000	−1.1; ±5.2	Triv000	−2.7; ±5.3	Triv000	
SJ mean power	−6.0; ±5.7	Triv00 (p < .05)	−2.5; ±8.4	Triv00	−4.8; ±9.9	Triv+	
SJ peak force	0.1; ±2.4	Triv0000	−0.3; ±3.4	Triv000	−1.3; ±3.1	Triv000	
SJ RFD max	−1.8; ±11.4	Triv0	−11.3; ±13.7	Small++	−10.5; ±15.0	Small+	
SJ duration	2.9; ±5.3	Small+	0.4; ±6.8	Trivuncl	1.7; ±7.5	Trivuncl	
MJ height	−5.1; ±8.5	Small+	−6.0; ±6.3	Small+	−1.7; ±5.3	Triv00	
MJ RSI	−3.5; ±9.0	Triv+	−4.2; ±6.5	Triv+	1.2; ±5.6	Triv000	
MJ vertical stiffness	−2.7; ±9.7	Triv+	10.8; ±12.4	Small**	4.0; ±10.7	Trivuncl	
20 m	1.4; ±1.0	Triv+	0.5; ±1.2	Triv00	−0.1; ±1.1	Triv000	
Push-up peak force	−0.3; ±7.8	Trivuncl	0.4; ±9.3	Triv00	−9.0; ±8.9	Small+ (p < .05)	
Squat peak power	–	–	−9.1; ±5.6	Small++ (p < .05∕01)	−8.8; ±5.5	Small++ (p < .05∕01)	
Bench press peak power	–	–	−5.3; ±4.7	Triv00 (p < .05)	−5.7; ±7.7	Triv+	
Squat estimated 1RM	–	–	1.1; ±3.1	Triv0000	−1.2; ±4.2	Triv000	
Bench press estimated 1RM	–	–	−1.6; ±3.4	Triv0000	−1.7; ±4.5	Triv000	
sRPE whole body	−20.6; ±10.1	Small+++ (p < .05∕01)	–	–	–	–	
sRPE upper body	−16.9; ±10.7	Small++ (p < .05∕01)	–	–	–	–	
sRPE lower body	−21.9; ±8.8	Small+++ (p < .05∕01)	–	–	–	–	
PRS whole body	−11.1; ±7.0	Small+ (p < .05∕01)	−10.1; ±9.2	Small+ (p < .05)	−6.8; ±10.4	Triv+	
PRS upper body	−9.4; ±4.8	Triv+ (p < .05∕01)	−10.2; ±8.8	Small+ (p < .05)	−10.1; ±10.9	Small+	
PRS lower body	−8.5; ±6.6	Triv+ (p < .05)	−12.8; ±11.2	Small+ (p < .05)	−6.1; ±10.8	Triv00	
Notes.

1RM 1 Repetition Maximum

CI Confidence Interval

MJ Multi Jump

PRS Perceived Recovery Status

RSI Reactive Strength Index

RFDmax Maximal Rate of Force Development

SD Standard Deviation

SJ Squat Jump

sRPE session Rate of Perceived Exertion

Trivial (Triv), <0.2, Small: 0.2-0.6; Moderate (Mod), 0.6-1.2; Large, 1.2-2.0; Very large, 2.0-4.0; Extremely large, <4.0 *, Possibly beneficial; **, Likely beneficial; ***, Very likely beneficial; +, Possibly harmful; ++, Likely harmful; +++, Very likely harmful; ++++, Most likely harmful; 0, Possibly trivial; 00, Likely trivial; 000, Very likely trivial; 0000, Most likely trivial; uncl, Unclar (need more data); p < .05, The 95% CI do not overlap with zero; p < .01, The 99% CI do not overlap with zero.

Discussion

In this study, we aimed to compare the recovery rates after a heavy-load, strength-oriented session and a moderate-load, power-oriented session of similar concentric work. Our main findings were: (1) The strength-oriented session had, overall, the largest detrimental effects on the neuromuscular system, impairing both the eccentric and concentric phases of jumping. However, the differences in performance assessments between the sessions were generally of small or trivial magnitudes. (2) The most sensitive recovery-markers for demonstrating reduced capacity and a difference between the strength-oriented session and the power-oriented session were CMJ eccentric and concentric peak forces, SJ RFDmax and squat peak power; these variables displayed small, but likely clear differences between sessions after 24 and 48 h of recovery. (3) In contrast to the strength-oriented session, the power-oriented session seemed to potentiate performance, as we observed small increases in MJ height after 24 h and in squat peak power after 48 h. (4) Finally, the strength-oriented session was perceived as more strenuous and the rate of recovery as slower compared to the power-oriented session; however, subjective and objective measurements correlated poorly at the individual level.

Figure 4 Objective vs. subjective measures.

X-y-plots of individual values for the strength-oriented session and the power-oriented session; regression lines are given with 95% confidence bands. A and B display the relationship between eccentric peak force and perceived recovery status (PRS; lower body) 24 and 48 hours after the sessions. C and D display the relationship between peak push-up force and PRS (upper body) 24 and 48 hours after the sessions. PRS values are given in the range 0–100, where 100 is fully recovered.

Previous studies

Small to trivial impairments of neuromuscular performance were observed after both the exercise sessions. More specifically, measures of CMJ and SJ heights and sprint times were reduced by 1–8%, which are at the low end compared to previous studies (∼2–20%; Raastad & Hallen, 2000; Howatson, Brandon & Hunter, 2016; Raeder et al., 2016; Davies, Carson & Jakeman, 2018; Hiscock et al., 2018). We believe that this discrepancy is because our participants were well trained, and more importantly, they were familiarized with the exercises and tests.

In line with the existing literature (Linnamo, Hakkinen & Komi, 1998; Brandon et al., 2015; Howatson, Brandon & Hunter, 2016), a heavy-load strength-oriented session attenuated the neuromuscular system more than a low or moderate load power-oriented session. However, in previous studies where the exercise work was controlled for, the differences between strength- and power-oriented sessions were close to eliminated (Mccaulley et al., 2009; Hiscock et al., 2018). Our observations confirm these findings, but add some nuances to this picture, as we observed some clear differences between the strength-oriented session and the power-oriented session, such as for CMJ eccentric peak force. Nevertheless, the differences in recovery rates between resistance exercise sessions of different modes (strength- and power-oriented) with similar exercise work must be expected to be rather subtle in magnitude, but these differences may still be relevant information and important for athlete monitoring and training planning. When small differences are of importance, we must, however, ensure that we have adequate measurement methods.

Methodological issues: reliability and fatigue sensitivity

To discriminate between the recovery rates of closely related exercise modalities such as strength- and power-oriented sessions, highly reliable (day-to-day) tests must be applied. Based on our familiarization session, and two pre-session tests, we observed very high reliability for the sprint test (CV: ∼1%). CMJ and SJ height and estimation of 1RMs had good reliability (CV: 3–5%), while peak power in the squat and bench press and MJ height had acceptable reliability (CV: ∼9–10%). Push-up peak force reached near acceptable reliability (CV: ∼11%). Overall, the reliability of tests applied in this study is in line with those of others (Raastad & Hallen, 2000; Hopkins, Schabort & Hawley, 2001; Byrne & Eston, 2002; Cronin, Hing & Mcnair, 2004; Cormack et al., 2008; Taylor et al., 2010; Gathercole et al., 2015a; Gathercole et al., 2015b). One exception among our tests was the RFDmax gleaned from CMJ and SJ, which demonstrated poor reliability (CV >20%). Previous studies confirm a moderate to poor reliability for RFD measurements in single joint knee-extension (CV = 7–17%) (Buckthorpe et al., 2012), and for CMJ and SJ (CV = 16–18%) (McLellan, Lovell & Gass, 2011; Gathercole et al., 2015a). The low reliability for RFD is probably related to the complexity of the task (Maffiuletti et al., 2016), meaning that it is more difficult to achieve a true maximal RFD than a maximal force. This seems to be reflected in studies showing larger increases in RFD than in maximal force within a session (rehearsal) and as an effect of training (Holtermann et al., 2007). That said, better reliability of RFD measures might be achieved by other types of tests than the jump tests applied here; indeed, the isometric mid-thigh pull test appears to be a preferable choice to assess RFD in the lower body (Haff et al., 2015; Hornsby et al., 2017).

Performance tests may also be evaluated by comparing the “smallest worthwhile change” (SWC) with the typical error (Cormack et al., 2008): If the SWC is larger than the typical error, the test should allegedly be able to (confidently) detect relevant and meaningful changes. Among our tests, jump height and measures of force (concentric and eccentric peak force) demonstrated CVs equal to or lower than the SWCs (see Table 2). Nevertheless, an evaluation of tests must be applied in practice. Gathercole et al. (2015a) used the term “fatigue sensitivity”, which refers to a test’s ability to detect impairments in neuromuscular function after exercise. As the conditions of the neuromuscular system change, due to different forms of central and peripheral fatigue (Enoka et al., 2011), high reliability measured in the rested state is not necessarily valid for the fatigued state. In fact, tests of isolated joints, such as isokinetic knee-extension assessments, appear to demonstrate larger changes than multi-joint tests, such as sprint and jump tests after different multi-joint activities (Byrne & Eston, 2002; Andersson et al., 2008; Howatson, Brandon & Hunter, 2016). To this end, we suggest that tests enabling subtle changes in the movement pattern, such as sprint and CMJ, may be highly reliable, but may lack fatigue sensitivity. Subtle movement/technique compensations that optimize the conditions for the current state of the neuromuscular system may indeed “mask” fatigue if only jump height in a CMJ is considered (Van Ingen Schenau et al., 1995; Gathercole et al., 2015b).

As indicated above, there were trivial changes in CMJ height and peak power 24 and 48 h after both sessions, but clear changes in CMJ eccentric time and CMJ eccentric peak force. Similar findings have recently been reported by others (Gathercole et al., 2015a). These observations indicate that the participants’ ability to use the eccentric phase was impaired in the recovery phase, but some compensations in the execution of the jump apparently minimized the reductions in jump height and power production. After the strength-oriented session the reduction in eccentric peak force seemed related to a slower eccentric phase during the CMJ; i.e., increased eccentric time, since the lowering the of centre of mass was not changed. On the contrary, the participants appeared to lower their centre of mass more after the power-oriented session than at pre-test, especially at 48 h. Future studies should investigate changes in the kinetics and kinematics (movement strategies) of a CMJ in the recovery phase compared to the rested state. However, we suggest the eccentric peak force is a more sensitive marker of fatigue and neuromuscular impairments than jump height and maximal power.

We found no clear meaningful differences between sessions or in the recovery rates between sessions for CMJ and SJ heights. This contrasts with observations by Byrne & Eston (2002), who reported that SJ height was reduced more and recovered slower than CMJ (and drop jump) height after a squat exercise session (10 × 10 repetitions at 70% of body weight). The discrepancy between findings may be related to more muscle damage in the study by Byrne & Eston (2002) than the present study—as indicated by a larger drop in performance (Paulsen et al., 2012). Moreover, studies have investigated various measures of RFD and observed that the impairment and recovery of RFD differ from maximal force (Penailillo et al., 2015; Farup et al., 2016). In our study, we extracted RFDmax from CMJ and SJ, and despite low reliability, we report small unclear and possibly clear differences between sessions at 24 and 48 h—in accordance with previous observations (Gathercole et al., 2015a). Thus, we recognize RFDmax values from jump tests as possibly fatigue sensitive, but we warn about high day-to-day test variability (as discussed above). Moreover, the reader should be aware that sampling frequency and the methods used for (concentric/eccentric) phase identification may affect the outcomes of SJ and CMJ analyses (Owen et al., 2014; Eagles et al., 2015). Hence, comparisons across studies must be made with caution.

From the force-velocity tests in bench press and squat we calculated peak power and estimated 1RM. The 1RM values had allegedly good reliability (CV<5% and CV<SWC), but contrary to the peak power, the 1RM values showed trivial changes after both exercise sessions. Although it has been suggested to be worth using (Jovanovic & Flanagan, 2014; Scott et al., 2016), force-velocity estimated 1RM appears to have limited value for monitoring small changes in recovery status; i.e., estimated (or predicted) 1RM tests appear to have low fatigue sensitivity. We applied ∼90% of 1RM as the heaviest load, which may have been too low to get an accurate estimation of 1RM in the squat, as observed by some (Banyard, Nosaka & Haff, 2017). For the bench press, however, ∼90% of 1RM should be adequate for precise 1RM estimations—at least in an unfatigued state (Jidovtseff et al., 2011; Garcia-Ramos et al., 2018a).

Mechanisms for neuromuscular recovery

Exercise-induced impairment of neuromuscular function and the following recovery phase are multifaceted (Lieber & Friden, 2002; Enoka et al., 2011; Paulsen et al., 2012). However, if we consider a particular exercise, such as the squat, and assume a constant range of motion (muscle lengthening/strain) and a given total exercise volume (sets × repetitions), the determining factors would be narrowed down to contraction/lengthening velocity and force. With the criterion of maximal effort (intention to move) in the concentric phase, velocity will be high and force low during light or moderate load power exercises, and vice-versa for heavy load strength exercises (cf. the force-velocity relationship (Huijing, 1998)). Higher concentric forces during the heavy load strength exercises will logically put more mechanical stress on the muscle tissue. However, high-force concentric contractions result in minimal muscle damage and a swift recovery of muscle function within 24 h (Jones, Newham & Torgan, 1989; Lee, Suter & Herzog, 1999; Carson, Riek & Shahbazpour, 2002). Thus, concentric work can probably only explain perturbations in neuromuscular function shortly after exercise (i.e., minutes to a few hours, as a result of metabolic factors; Allen, Lamb & Westerblad, 2008). This led us to suggest that the eccentric phase was probably of greatest importance in the differences in neuromuscular impairment and recovery rates between sessions (Paulsen et al., 2012). In other words, the higher eccentric forces—simply due to higher loads—during the strength-oriented session likely explain the slower recovery compared to the power-oriented session (Faulkner, Opiteck & Brooks, 1992; Black et al., 2008). On the other hand, the between-session differences displayed by the recovery markers were generally small and trivial compared to the significant difference in loads (the loads in the power session were 50% of those in the strength session). Therefore, we propose that the higher eccentric velocity during the power-oriented (compared to the strength-oriented session) caused a substantial mechanical stress on the muscles, despite the moderate loads: Stretch-shortening cycle exercises have, indeed, been shown to induce muscle damage and require days of recovery (Nicol, Avela & Komi, 2006). Future research should investigate this, but we suggest that how the eccentric phase during power-oriented exercise is performed and the utilization of the stretch-shortening cycle could have a major impact on recovery times.

Lower and upper body exercise

In the present study, both upper body and lower body exercises were applied. Studies exploring muscle damage and recovery after eccentric exercise have reported that upper body muscles sustain more damage and require longer recovery times than lower body muscles (Jamurtas et al., 2005; Chen et al., 2011; Chen et al., 2019). However, recovery rates after traditional strength training do not appear to be different between upper and lower body exercises, such as the bench press and squat (Mclester et al., 2003; Korak, Green & O’neal, 2015; Moran-Navarro et al., 2017). In line with these studies, our data demonstrate a similar recovery rate for upper and lower body exercises. Moreover, as for the lower body, the strength-oriented session seemed to induce somewhat more fatigue and longer recovery times than the power-oriented session for the upper body. In contrast to most studies that have investigated recovery after eccentric exercise (as cited above), we recruited well-trained individuals, which points to training status as an important parameter for recovery times—rather than an inherent difference between upper or lower body muscles. Nevertheless, great care should be taken when comparing recovery from different exercises/sessions, because variables such as muscle strain, force and work are very difficult to control for.

Fatigue vs. potentiation and supercompensation

Neuromuscular function can be altered through adaptation to training over weeks and months (Goldspink, 1985), but the neuromuscular system is also history-dependent for shorter time periods. In fact, both fatigue and potentiation are possible outcomes of muscle contractions (Sale, 2002). While heavy loads and large exercise volumes may induce long-lasting neuromuscular fatigue (hours and days), exercises conducted with low volume and high/maximal effort can result in potentiation and enhanced neuromuscular function that lasts for minutes to several hours (Cook et al., 2014; Russell et al., 2016). Interestingly, in the present study the power-oriented session appeared to enhance MJ height at 24 h and squat peak power and push-up peak force 48 h after exercise (note that the push-up peak force at 48 h was trivial and unclear compare to baseline, but clearly different between sessions). This is in line with Tsoukos et al. (2018), who observed increased CMJ height and RFDmax 24 and 48 h after loaded jump squats (40% of 1RM; 5 × 4 repetitions). In contrast to squat peak power, we observed no such “supercompensation” in CMJ, SJ or 20 m sprint (which were all back to baseline at 48 h). Notably, our participants executed a large exercise volume, about three times that of Tsoukos et al. (2018), and fatigue mechanisms may have overshadowed most of the supercompensation effects of power exercises. Moreover, we only followed the participants for 48 h, which means that we do not know whether the supercompensation occurred later after the strength-oriented session (e.g., after 72 h). As final note, potentiation effects (or supercompensation) is indeed relevant for athletes, as it is common practice for “power athletes”, e.g., rugby players, track and field throwers and sprinters, to perform a power-oriented session close to competitions (∼4-48 h; Russell et al., 2016; and own observations from the Norwegian Olympic Center, Oslo, Norway).

Objective vs subjective measures of recovery

Session RPE (sRPE) for resistance exercise was reviewed by Mcguigan & Foster (2004) and validated for “intensity”; i.e., load in % of 1RM, by Sweet et al. (2004). Later studies have found the sRPE to be related to both volume and work rate during strength training (Scott et al., 2016; Hiscock et al., 2018). The present study ensured equal concentric work, but different loads—i.e., the power-oriented session was performed with 50% of the loads used in the strength-oriented session. Nevertheless, because the power-oriented session lasted ∼12% (∼13 min) longer than the strength-oriented session, the work rate was highest during the strength-oriented session. As the difference in loads (% of 1RM) between sessions was much larger than the difference in work rate, we suggest that the higher loads (% of 1RM) were the dominant factor influencing the sRPE scores (although we acknowledge that this cannot be ascertained with the present study design). Notably, it has been proposed that exercise intensity/load (% of 1RM) influences RPE scores via a positive relationship with the central motor control discharge (Gearhart Jr et al., 2002), cf. the “corollary discharge model” (Pageaux, 2016). However, our participants in both sessions were strongly encouraged to execute every repetition with the intention to move as fast as possible in the concentric phase. Indeed, both the motor-related cortical potentials (MRCP; Slobounov, Hallett & Newell, 2004) and the electromyographic (EMG) amplitude seem independent of load (% of 1RM) if the intention to move is maximal—at least for lower body exercises (Bosco et al., 1982; Hakkinen, Komi & Kauhanen, 1986; Kawamori & Haff, 2004; Mcbride et al., 2010). If we assume that our participants moved maximally in all repetitions, the corollary discharge model seems unable to explain a higher sRPE after the strength-oriented session than the power-oriented session. Consequently, we suggest that the sRPE scores in the present study were influenced by afferent feedback from the muscles; supporting a “combined model” (Pageaux, 2016). The afferent feedback may be a combination of different sensors including tendon organs (“force sensors”) and nociceptor receptors responding to metabolic perturbations. Metabolic perturbations, such as elevated extracellular levels of adenosine, lactate and protons (Allen, Lamb & Westerblad, 2008), stimulate capsaicin fibres (Aδ and C-nerves; Pollak et al., 2014); and accordingly, muscular fatigue may be an important underlying mechanism behind the RPE scores (Hardee et al., 2012; Vasquez et al., 2013). When working at maximal intensity, fatigue will start to develop within seconds (Allen, Lamb & Westerblad, 2008), and probably to a larger degree during the strength-oriented session than the power-oriented session due to more time under tension (i.e., a longer acceleration phase during the lifts and/or less deacceleration). We cannot exclude the possibility that the participants used elastic energy storage and release (the stretch shortening cycle) during the power-oriented session, and thereby had better energy economy during the power-oriented session than the strength-oriented session (Bosco et al., 1982). Higher energy expenditure and more fatigue in combination with the heavier loads could explain the higher sRPE after the strength-oriented session than the power-oriented session. Finally, it is noteworthy that the “contents”/definition of the RPE concept, i.e., effort vs. force, pain and discomfort, and the mechanisms behind RPE, are debatable (Pageaux, 2016). Moreover, the timing of reporting RPE, e.g., during or immediately after an exercise vs. 30 min after a session (i.e., sRPE), may be important for the decisive mechanisms of the RPE scores; thus, more scientific work is needed to better understand the use of sRPE in relation to different modes of resistance exercise.

While sRPE scores are collected after a session, PRS is obtained before an exercise session. PRS is supposed to give an evaluation of the athletes’ readiness and performance status in the upcoming session (Laurent et al., 2011). In the present study, recovery status 24 and 48 h after the strength-oriented session were reported lower compared to the power-oriented session. Indeed, as for sRPE, PRS pointed in the same direction as the objective tests. However, no consistent correlations were found between the PRS and objective variables, such as CMJ eccentric peak force and push-up peak force. Interestingly, the state of recovery was perceived as incomplete both 24 and 48 h after the power-oriented session although performance was back to baseline, or even above (squat peak power and MJ). Recent studies support a partly dissociated time course between objective and subjective recovery status—for both upper and lower body muscles—indicating a slower recovery when assessed subjectively (Zourdos et al., 2016; Ferreira et al., 2017a; Ferreira et al., 2017b; Marshall, Cross & Haynes, 2018). In summary, this advocates for caution in interpreting subjective and objective measures of recovery. In our case (and perhaps most cases), it is conceivable that neither the subjective nor the objective measures revealed the true recovery status. On the objective side we merely measured some properties of the neuromuscular system, leaving the possibility that unassessed properties were not recovered. Interestingly Zourdos et al. (2016) observed a difference in the PRS when assessed before and after warm-up (higher PRS after warm-up). We assessed PRS only before warm-up, leaving the possibility for higher coherence between objective measurements and PRS if evaluated after warm-up.

Limitations

The present study has limitations. First, we applied a series of tests and we cannot exclude the possibility that the tests themselves induced fatigue that affected the results; e.g., reduced the test reliability. Moreover, we had no control trial in which the participants simply conducted the four test-sessions without participating in an exercise session (see Fig. 1). Consequently, we must be careful interpreting the changes in relation to time after each session (within-session changes); it is possible that the recovery was prolonged due to all the tests.

Second, we calculated the work done based on concentric work; thus, we excluded eccentric work, and we cannot rule out that some differences between sessions could have been explained by this fact.

Third, each participant completed two sessions. Due to the repeated bout effect, a faster recovery must be expected after the second session (Mchugh, 2003). Moreover, since the loads (in % of 1RM) were higher in the strength-oriented session, the adaptative processes may have been better stimulated after the strength- than the power-oriented session (i.e., strengthening of the myofiber cytoskeleton (Paulsen et al., 2009)). If true, this may have created a bias toward faster recovery after the power-oriented session. Furthermore, the time between sessions (the washout period) varied between the participants (1–4 weeks), which means that their training status may have changed slightly. This effect does appear small as the pre-values before each session were very similar, with low to moderate CV for all variables (Table 2). To this end, the order of sessions was randomized, and the impact of session-order was trivial when controlled for. Additionally, we recruited both females and males. The participants had different training backgrounds and we did not control their training in the washout period (except during the 48 h before each session). We did not fully control the diets of the participants. We acknowledge that these factors may have induced biases and variability in our results.

Fourth, we did not include tests that allowed us to distinguish between central and peripheral fatigue, nor did we measure systemic markers of recovery (such as creatine kinase, testosterone and cortisol; Buckthorpe, Pain & Folland, 2014; Hiscock et al., 2018; Tsoukos et al., 2018). This could have given us valuable information about the subtle impairments of neuromuscular performance and recovery between sessions.

Fifth, we acknowledge that the definition of the different variables gleaned from the SJ and CMJ tests are open for debate. Particularly, we want to make the reader aware of the fact that CMJ peak eccentric and concentric force are reached within a very narrow time window in the lowest position of the jump. Thus, collecting only the force in the lowest squat position could yield the necessary information, with the advantage that the point/position is clearly defined (easily reproducible).

Finally, one should be careful about extrapolating the results of this study to other training interventions/programs due the many combinations/possibilities within a strength/power training program that may be important in the recovery process, such as exercises, load, volume, work and interest-rest periods.

Practical applications

Knowledge of recovery from exercise sessions is needed to make qualified assumptions when designing training programs, particularly for elite athletes who must handle large training volumes and avoid overtraining. The present and previous studies have shown that to monitor recovery one must consider a combination of tests and be aware of the error of measurements. In our study, the eccentric peak force during a CMJ and the peak power calculated from a squat force-velocity test were the variables that seemingly best differentiated between a strength-oriented session and a power-oriented session. Further research is warranted to see whether these tests are valid for other modes of resistance exercise and with participants of different performance levels (training status).

In our hands, RFD from CMJ and SJ seem to have too large a day-to-day variability to be recommended for monitoring recovery. Improved standardizations and instructions to the athlete may be worth exploring. Similarly, for the upper body our applied tests were not fully satisfactory in terms of reliability and fatigue sensitivity, implying that more work is needed.

The power-oriented session tended to improve performance in certain tests at 24 and/or 48 h after exercise. Potentiation or a fast supercompensation from power-oriented sessions is highly relevant for athletes preparing for competitions.

Objective and subjective tests of recovery may not correlate. Consequently, both test modalities should be used and interpreted together to ensure a holistic approach (Kiely, 2012). Because the recovery process is so complex, it is important to acknowledge that there is much we do not know or understand; thus, relying on only objective or only subjective measurers could prove inadequate for most athletes.

It appears that the best tests for assessing recovery will differ significantly according to the exercises that have been conducted. Consequently, we cannot expect a “gold standard” test battery. Rather, we need to use a selected number of tests for each specific athlete or group of athletes, and a combination of subjective and objective tests appears advisable.

Conclusion

We hypothesized that a heavy-load, strength-oriented exercise session would require a longer recovery period than a moderate-load, power-oriented session with equal concentric work. Our hypothesis was confirmed as the power-oriented session required less than 48 h of recovery, while the strength-oriented session required more than 48 h. The strength-oriented session induced an overall larger detrimental effect on the neuromuscular system than the power-oriented session at all time points (0, 24 and 48 h), reducing both power and strength properties. However, differences in the performance assessments between the exercise sessions were generally small or trivial. The apparently best markers for detecting differences between the strength-oriented session and power-oriented session were the CMJ derivate eccentric peak force and squat peak power. Considering the good reliability (lower than the SWC), the CMJ eccentric peak force seemed to be the most sensitive parameter. For the upper body, the push-up peak force seemed more sensitive as a recovery marker than bench press peak power and 1RM, but the push-up had only acceptable reliability. In contrast to the strength-oriented session, the power-oriented session seemed to potentiate multi-jump performance and squat peak power. Furthermore, the strength-oriented session was experienced as more strenuous (higher sRPE) and more recovery was perceived to be required (lower PRS) compared to the power-oriented session, which were in accordance with our secondary hypothesis. However, our third hypothesis was falsified as the subjective measurements correlated poorly (inconsistently) with the objective measurements; indicating the need for both objective and subjective measurements in practice.

Supplemental Information

Data 1 Individual data

Click here for additional data file.

We would like to thank all the participants for their hard work, the staff at the Norwegian Olympic Sports Center for good help facilitate the testing and training, and Biomekanikk AS for the help with processing force plate data.

Additional Information and Declarations

Competing Interests

Author Contributions

Ethics

Data Availability

Author Daniela Schäfer Olstad was employed by the company Polar Electro Oy. Of note, the present study does not contain any data collected by Polar Electro Oy equipment/devices. The remaining authors declare that they have no competing interests.

Christian Helland and Gøran Paulsen conceived and designed the experiments, performed the experiments, analyzed the data, prepared figures and/or tables, authored or reviewed drafts of the paper, and approved the final draft.

Magnus Midttun performed the experiments, analyzed the data, prepared figures and/or tables, and approved the final draft.

Fredrik Saeland performed the experiments, prepared figures and/or tables, and approved the final draft.

Lars Haugvad conceived and designed the experiments, performed the experiments, prepared figures and/or tables, and approved the final draft.

Daniela Schäfer Olstad conceived and designed the experiments, analyzed the data, prepared figures and/or tables, authored or reviewed drafts of the paper, and approved the final draft.

Paul Andre Solberg analyzed the data, prepared figures and/or tables, authored or reviewed drafts of the paper, and approved the final draft.

The following information was supplied relating to ethical approvals (i.e., approving body and any reference numbers):

The study was reviewed by the Norwegian Regional Ethical Committee of Medical and Health Research (approval number: 2016/1120).

The following information was supplied regarding data availability:

The raw data set is available in the Supplemental Files.

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
