# Peer review of "A strength-oriented exercise session required more recovery time than a power-oriented exercise session with equal work"

_PeerJ, doi:10.7717/peerj.10044_

## Round 0.1 · original submission · Major Revisions

Reviewers generally provided positive feedback regarding the impact of the study. However, they also revealed many areas in which the manuscript needs to be improved. The manuscript requires extensive revision and the quality of the English level is one of the aspects that need to be improved. I look forward to receiving a revised version of the manuscript.

·

Basic reporting

The authors should be commended for this. Although minor revisions are needed, the basic reporting of information is very clear and progressive and uses the literature well to support key points.
One relatively major criticism though focuses on a lack of detail in key elements of the method section, particularly with regards to how force data were processed (I feel that more detail is needed) and the use of MBI statistics given recent criticism (please justify your use of this in the face of the robust criticism).

Experimental design

I think that the authors should be commended on their experimental design that is both rigorous and appropriate for the question they try to answer.
This appears to be a technically high quality study, but far more detail is required in key elements of the method section that, when addressed, would enable researchers to replicate and build on this work (please see previous section).

Validity of the findings

I think that the authors discuss their results well, but while the points they make and the conclusions they reach appear robust and meaningful, I also think that thread of the story that is been told could be clearer and that this is largely a consequence of the way the discussion has been organised - I feel that this could be both clearer and more concise but I also understand that I could be biased because of how I approached the discussion section during review.

Additional comments

Thank you for giving me the opportunity to review this manuscript! I feel that the authors ask a very relevant question and use appropriate experimental design and methods to answer this. I think that their results clearly contribute to the existing literature and will be practically useful/applicable.
Please see the attached, annotated PDF for more specific feedback.

Reviewer 2 ·

Basic reporting

Unfortunately, the article doesn't make sense in many places. Thus, the authors need to edit the English and then have this reviewed.

Experimental design

It appears to be OK but the English needs to be improved.

Validity of the findings

These appear to be OK but the English needs to be improved.

Additional comments

Thank you for the opportunity to review this manuscript. I would like to note that that the question is warranted and interesting and I commend the authors for the use of MBIs. However, please do note that it has now changed to MBDs (refer to Hopkins Sportsci website for further information).
Unfortunately, the writing of this manuscript is not up to standard and after reading just the first few paragraphs it is clear that it would not be appropriate to accept this until it is edited and the English is improved. I understand that English mightn’t be the authors first language and this does put you at a disadvantage, but this is an English speaking journal and we must abide by the rules. It is not the job of the reviewer to edit every sentence of a submitted manuscript, it is our job to uphold the scientific rigour. Therefore, for me to spend the whole time editing the English is not fair on us as reviewers. To help you (and demonstrate the magnitude of changes that are required), I have provided comments for the first paragraph (and abstract) below.
Finally, I would warn the authors against using the terms a “power session” or a “strength session”. While I understand what the authors are getting at, power can be developed through training at either intensity and so can strength (this is well-documented throughout the literature). Therefore, I would strongly advise calling them a “power-oriented” and “strength-oriented” session.

Line 30 should use “greater” rather than “more”. The word “more” doesn’t make sense in this context.
Sentence starting line 31 “In accordance…” (in particular: “…was experienced more strenuous…”) doesn’t make sense. Please fix this.
Sentence starting line 34: “In conclusion….” You are missing the word “a” after “observed” and before “larger”

Line 45, first sentence: Doesn’t make sense. Please fix this.
Sentence starting line 47: this is far too vague. The reader isn’t clear what adaptation processes you are referring to.
Sentence starting line 50: “However, our knowledge recovery processes….” This sentence doesn’t make sense.

Sentence 52: “based on the existing literature….”. This sentence is far too colloquial for an academic text.
Line 57: “recovery time levels off at a certain volume” It isn’t clear what you are meaning.

Line 59: “Moreover, when lifting weights (“isotonic” muscle work) we can expect longer recovery times with increasing relative loads; possibly as a consequence of the correspondingly higher eccentric force-generation (Faulkner et al., 1993; Black et al., 2007;Raeder et al., 2016;Hiscock et al., 2018).”
If this is the case, then why are you doing the study? You have already answered your own question.

The sentence below doesn’t make sense:
Long-lasting recovery (days) of the neuromuscular functions can largely be explained by damage and disturbances in the excitation-contraction-coupling and the myofibrillar machinery (Paulsen et al., 2012), although central (neural) fatigue may persist for some time (Nicol et al., 662006;Enoka et al., 2011;Carroll et al., 2017).

Reviewer 3 ·

Basic reporting

Overall, English language usage is clear and unambiguous. However, authors should improve the density of the writing, being more straight forward and constrained to the scope of the problem and data at hand. For instance, the Intro presents a lit review, instead of the problem at hand, and the Discussion is way too speculative.

Literature references, field background and context are over provided. Even though with no major flaw, paper is not reading well as you are making excessive extrapolations based on the data that authors collected. For instance, you are talking about muscle damage, central fatigue, etc.

Experimental design

Cross-over designs cannot be implemented as completely randomized designs. Session order needs to be counterbalanced between subjects to mitigate the carryover effect, characteristic of cross-over designs. This is an important limitation of the study. Besides, the time interval (1-4 wks) between training sessions may be an important confounding factor as participants may change their training state.

The characteristics of the sample (different training backgrounds and the small sample size) may introduce important bias and variability to the findings. For instance, you have a few volleyball players, a couple bikers and physical education students.

How did you control for training load outside the study as some participants were athletes that should have trained at least a few days per week throughout the experimental protocols. Besides training background can directly affect the recovery rate and duration and you did not present any sort of control for this confounding factor.

Validity of the findings

As exercises, volume and intensity were arbitrarily selected, extrapolations to other strength and power training sessions are rather difficult.

Even though a group of sport scientists have supported the use of magnitude-based inference, a few statisticians have written papers severely criticizing this method and clearly stated that it should be abandoned. A more traditional approach would be welcome (standard inferential analyses and effect sizes). Authors should present the rational for presenting only a few correlations.

Conclusions are OK, but including RDFmax as a marker of fatigue is unreasonable as the reliability of the measurement is poor.

The introduction and discussion sections have way too much speculation and should be removed.

Annotated reviews are not available for download in order to protect the identity of reviewers who chose to remain anonymous.

---

## Round 0.2 · Minor Revisions

Authors did a good job replying to the reviewer comments. Unfortunately, I did not hear back from previous reviewers and I performed an analysis of the paper and I have provided several additional specific comments to the authors. Please, consider them and I am looking formard to receiving a revised version of the manuscript.

Lines 20-21. Change to “The present cross-over controlled study aimed to compare the rate of recovery between strength-oriented and power-oriented resistance training sessions with equal work.” Delete through the manuscript the words “heavy” and “moderate” before strength-oriented and power-oriented sessions, respectively.

Line 25. “replace “perceived rate of exertion (RPE)” “rate of perceived exertion (RPE)”.

Line 27. Add “decrements in all variables” at the end of the sentence if this is the case.

Line 34. Delete the word “session”.

Lines 143-148. Hypotheses should be more specifically defined and supported by citations. Clearly indicate what it is expected for (I) mechanical variables, (II) perceptual variables, and (III) relationship between mechanical and perceptual variables. Keep in mind that the hypotheses should be confirmed/rejected by the statistical analysis. For example, it is not clear what is “a faster recovery”. It would be clearer to say that strength-oriented would show a higher “fatigue” that power-oriented at all time points, whereas strength-oriented are not expected to recover baseline values after 48 hours and power-oriented in expected after 24 hours. Whatever is the hypothesis of the authors but hypotheses should be less ambiguous.

Line 160. “equal volumes” or “equal work”?

Line 199-200. Delete “i.e., to perform a plyometric movement”.

Line 225. Best attempt according to which variable? Jump height? Please, specify.

Jump analysis. I am curious why 98.5% and 101.5% of body mass were used as the jump initiation thresholds. Any reference supporting this procedure? Specify also the threshold used to identify the point of take-off?

Line 288-289. Change to “Peak power was calculated as the apex of the parabolic power-velocity relationship”. How was estimated the 1RM? Does the software estimate the 1RM from the individualized load-velocity relationship (https://journals.lww.com/nsca-scj/Abstract/9000/Velocity_Based_Training__From_Theory_to.99257.aspx)? Explain procedure used to estimate the 1RM.

Line 317. It is not clear which was the purpose of measuring velocity. It was not used it should be deleted because it may create confusion.

Line 536. As observed in the squat (Banyard et al.,) but not in the bench press (https://pubmed.ncbi.nlm.nih.gov/28872384/). Using the 90%1RM it can be obtained a highly accurate estimation of the bench press 1RM. In fact in this study, although it is not clear the procedure used to estimate the 1RM (it should be better explained), the 1RM was obtained with a high reliability.

---

## Round 0.3 · accepted · Accept

Given that you were interested in the changes in maximal strength ("1RM"), it could be acceptable the the use of musclelab software for "1RM" estimation. However, my recommendation is that in the future you estimate the 1RM using more precise methods as I indicated in my previous review. This procedure to estimate 1RM through L-V relationship is quite simple and it will help our community to have more standard measures and will allow to compare results between different studies. Despite this suggestion, I believe authors conducted a really nice study which deserves publication in PeerJ.